# Dysregulation of Natural Killer Cells in Obesity

**DOI:** 10.3390/cancers11040573

**Published:** 2019-04-23

**Authors:** Donal O’Shea, Andrew E. Hogan

**Affiliations:** 1Department of Endocrinology, St. Vincent’s University Hospital & University College Dublin, Dublin 4, Ireland; info@dosheaendo.ie; 2National Children’s Research Centre, Crumlin, Dublin 12, Ireland; 3Human Health Institute, Maynooth University, Co. W23 F2K8 Kildare, Ireland

**Keywords:** cancer, NK cell, obesity, metabolism, adipose tissue

## Abstract

Natural killer (NK) cells are a population of lymphocytes which classically form part of the innate immune system. They are defined as innate lymphocytes, due to their ability to kill infected or transformed cells without prior activation. In addition to their cytotoxic abilities, NK cells are also rapid producers of inflammatory cytokines such as interferon gamma (IFN-γ) and are therefore a critical component of early immune responses. Due to these unique abilities, NK cells are a very important component of host protection, especially anti-tumour and anti-viral immunity. Obesity is a worldwide epidemic, with over 600 million adults and 124 million children now classified as obese. It is well established that individuals who are obese are at a higher risk of many acute and chronic conditions, including cancer and viral infections. Over the past 10 years, many studies have investigated the impact of obesity on NK cell biology, detailing systemic dysregulation of NK cell functions. More recently, several studies have investigated the role of NK cells in the homeostasis of adipose tissue and the pathophysiology of obesity. In this review, we will discuss in detail these studies and focus on emerging data detailing the metabolic mechanisms altering NK cells in obesity.

## 1. Obesity

Obesity is classically defined as an accumulation of excess adiposity. The current strategies for stratifying bodyweight are based on body mass index (BMI) which measures bodyweight in kilograms and divides by height in metres squared. A BMI between 20 and 25 kg/m^2^ represents a healthy bodyweight, with a BMI between 26 and 30 defining overweight, and a BMI greater than 30 classified as obese. Over 600 million adults worldwide are classified as obese, with an estimated age-standardized prevalence of 10.8% among men and 14.9% among women [1]. For the first time, more children are obese than underweight. This represents a 10-fold increase in childhood obesity since 1975 [2]. The estimated age-standardized prevalence of childhood obesity is 5% [3]. We know now that childhood obesity tracks strongly into adulthood, with up to 82% of obese children remaining obese in adulthood [4]. Large population studies have demonstrated that obesity increases the risk of developing numerous comorbidities, including type 2 diabetes mellitus (T2DM), cardiovascular disease (CVD) and increased risk of many cancers [5,6]. These co-morbidities have been clearly shown to reduce the life expectancy of the obese individual. 

## 2. Obesity and Cancer

Cancer is a leading cause of death worldwide with 7.6 million deaths from cancer annually [7]. Recent reports from the centers for disease control and prevention (CDC) in the United States have implicated obesity in 40% of cancer cases, representing 630,000 diagnosis in 2014 alone [8]. This is a finding which is mirrored in many studies around the globe [9]. The development of obesity related cancers is multi-factorial, with many changes in hormones, adipokines, cytokines and immunity identified. Obesity is associated with insulin resistance, a well-established pre-cursor to T2DM; however, elevated insulin levels also promote the development of certain obesity related cancers such as colorectal, pancreatic, liver and endometrial [10,11,12]. Several studies have also identified links between leptin, an adipokine increased in obese children and adults, and cancer. Most studies investigating leptin in the context of obesity have focused on breast cancer, where leptin can directly stimulate the proliferation of breast tumour cells [13,14]. Another important component of cancer development in obesity is the presence of chronic inflammation [15,16]. Numerous studies have highlighted the links between chronic inflammation and the development of cancer [15]. Amongst the strongest associations between inflammation and cancer is in colon carcinogenesis in patients with inflammatory bowel disease [17]. 

## 3. Obesity and Immune Dysregulation

Obesity drives a programme of systemic inflammation, which underpins the development of not only malignancies, but other co-morbidities such as T2DM and CVD. Several cytokines which are elevated in obese individuals have been shown to disrupt normal homeostatic processes, such as interleukin 1 (IL-1) and insulin signalling, or IL-17 and adipogenesis [18,19]. Obesity driven inflammation has been shown to promote the development of liver cancer [20]. This chronic inflammation stems primarily from a dysregulated immune system. We and many others have identified alterations in subsets of the immune system, paired with a loss of regulation and increased inflammatory profile [21,22,23,24,25,26,27]. This dysregulation is not limited to adults with established co-morbid diseases, but is also present in obese children as young as six years of age, long before the development of overt disease [28,29]. Supporting the fact that obesity drives chronic inflammation, which precedes the development of co-morbid diseases including cancer. In addition to the increased inflammatory burden, obesity negatively impacts on the bodies anti-tumour effector populations, such as Natural Killer (NK) cells.

## 4. Natural Killer Cells

Natural killer cells are a subset of innate lymphocytes, which play a very important role in early host protection [30]. They differ from classical T and B lymphocytes in that, NK cells can rapidly respond to infected or transformed cells without prior activation, via their production of lytic molecules such as perforins or granzymes [31]. Natural Killer cells can also shape subsequent immune responses through their rapid production of cytokines (interferon gamma (IFN-γ), tumor necrosis factor alpha (TNFα), IL-6 and granulocyte-macrophage colony-stimulating factor (GM-CSF)). Activation of NK cells is tightly controlled through the expression of germline coded receptors which in response to changes in environmental cues can alter the balance of activating and inhibitory signals, allowing the activation of the cell. Loss of self-identifying molecules such as human leukocyte antigen (HLA) which provide an inhibitory signal via killer-cell immunoglobulin like receptors (KIRs) will result in the activation of NK cells, and the lysis of the target cell. Natural Killer cells can also be activated by the binding of activation molecules such as NKG2D which are upregulated by stressed and transformed cells [32,33]. They are therefore primed to target transformed tumour cells without prior immunological priming. In addition to their targeting of transformed cells, NK cells can also target cancer stem cells (CSC) which can seed metastasis and promote tumour bulk [34,35,36,37]. Activation of NK cells is not limited to the recognition of malignant cells or CSC; NK cells can also be activated via soluble cytokines including IL-2, IL-12, IL-15 and IL-18 [38,39,40,41]. Immunometabolism has been highlighted as a critical component of NK cell activation, at rest NK cells metabolize glucose via glycolysis coupled to oxidative phosphorylation, yielding high levels of energy. Upon activation, NK cells rapidly increase their rates of aerobic glycolytic metabolism providing the biosynthetic precursors for cytokine and lytic granule production. Clear evidence for the importance of glycolysis for NK cell effector function is demonstrated by the inhibition of glycolysis using 2-dexoy glucose (2DG), which results in inhibition of NK cell cytokine production and lytic molecule production [38,42,43]. Mammalian target of rapamycin (mTOR) has been highlighted as a master regulator of NK cell metabolism; its requirement has been demonstrated to be essential for NK cell effector function [38,42]. Traditionally, NK cells were described as lacking memory; however, several studies have now shown that murine NK cells can form memory [44]. In a series of very elegant studies human NK cells have been shown to under training, where an initial cytokine stimulation results in a boosted response several days or weeks later [45,46]. This has opened up a new therapeutic avenue for the use of trained NK cells as a potent immunotherapy. 

## 5. Natural Killer Cells in Cancer 

NK cells are a critically important component of anti-tumour immunity, as they are equipped with potent anti-tumour machinery including cytotoxic granules, pro-inflammatory cytokines and death-inducing ligands such as FAS ligand and TNF-related apoptosis-inducing ligand (TRAIL). The importance of NK cells in protection from cancer is highlighted by the increased prevalence of cancer in humans with defective NK cells such as Chediak–Higashi disease, and validated by the exacerbation of cancer in NK cell deficient mouse models [47,48]. Studies have also highlighted associations between low NK cell activity and cancer risk. In an 11-year longitudinal study, it was demonstrated that individuals with low NK cell activity levels were at a higher risk of developing cancer, highlighting the importance of NK cells in the prevention of overt malignancy and immunosurveillance [49]. Similarly, in a cohort of 872 patients, individuals with low NK cell activity (IFN-γ production) had a 10-fold higher rick of colorectal cancer than those with high NK cell activity [50]. Conversely, in a cohort of colorectal patients and matched controls, expression of the high NK cell activity haplotype of NKG2D, HNK1, was associated with decreased risk of colorectal cancer [50]. In parallel with cancer risk, several studies have highlighted the importance of NK cells in the prognosis of patients with cancer [51,52,53]. In a cohort of 50 patients with primary squamous cell lung carcinoma, increased tumour infiltrating NK cell frequency was correlated with improved prognosis [51]. Similarly, in a cohort of patients with colorectal carcinoma, high NK cell infiltration into the tumour was associated with a favourable outcome [52]. Cancer can also induce defects in NK cells, using a *KRAS* mutation model of pancreatic cancer, Kaur and colleagues highlighted defects in NK cell frequencies and functions during the pre-neoplastic stage of pancreatic cancer, suggesting that cancer induces early defects in NK cells which allows the progression and expansion of the disease. In the same study, diet induced obesity compounded the alteration of NK cells in the tumour microenvironment, which promoted the expansion of pancreatic tumours [54]. This is supported by studies which investigated NK cell functions in patients with acute myeloid leukaemia (AML) which reported that NK cells were dysfunctional with reduced cytokine production and degranulation. Incubation of NK cells isolated from healthy donors with AML blasts induced similar defects suggesting the reported defects were cancer induced [55]. Mamessier and colleagues expanded on this concept by showing that human breast cancer cells could promote tolerance in NK cells by reducing the expression of activating receptors including NKG2D, and that tumour derived factors such as transforming growth factor (TGF)-β1 directly reduced NK cell activity [56]. Due to their potent effector functions, NK cells represent an attractive target for cancer immunotherapy. Several approaches have been highlighted, ranging from adoptive transfer of NK cells into patients, cytokine therapy or targeting/modifying NK cells receptors to boost their cancer killing function [57]. Adoptive transfer of NK cells involves the isolation of either autologous or allogenic NK cells, followed by their expansion and activation before transfusion in the patient [58,59,60,61]. Studies investigating autologous transfer have not demonstrated clinically relevant responses [59,60]. This is due to the inability of adoptively transferred cells to lyse tumour cells unless restimulated in vitro [60]. The allogenic approach has yielded more positive results, especially with KIR-mismatch in patients with haematological cancers such as AML [58,61]. In a 2010 clinical trial of adoptive transfer of allogenic NK cells, Rubnitz and colleagues presented striking data in childhood AML, showing 100% remission rates [62]. Another approach which has shown promise it the transfer of cytokine-trained NK cells, this involves harnessing the training or memory potential of NK cells. Isolated NK cells (autologous or allogenic) are activated in vitro using a cocktail of cytokines (IL-12, IL-15 and IL-18) and, after restimulation, show boosted effector responses [46]. In a phase-I clinical trial, cytokine trained NK cells demonstrated increased effector functions and a clinical response in five out of nine patients with four complete remissions [63]. Collectively, these studies show the critical role for NK cells in tumour immunity and their therapeutic potential in humans.

## 6. Natural Killer Cells in Obesity. 

In 2009, we reported that NK cells were altered in a cohort of severely obese patients [64]. We showed that NK cell frequencies were reduced in obese patients when compared to lean counterparts, along with alterations in the surface repertoire of NK cell activating and inhibitory molecules. In the same cohort of patients, NK cell cytotoxic capabilities were reduced, with NK cells isolated from obese patients killing significantly fewer of the K562 myeloid leukaemia cell line compared to healthy controls. Several other studies have also highlighted the negative impact of obesity on NK cells [22,65,66,67,68,69,70]. In a 2017 study, Viel and colleagues reported increased expression of activation markers such as CD69 on NK cell from obese patients in parallel with a loss of function as measured by degranulation and the production of macrophage inflammatory protein (MIP)-1β or IFN-γ [70]. NK cells resident in tissues are also impacted by obesity, in 2017, Shoae-Hassani and colleagues showed that NK cells resident in the adipose tissue of obese individuals expressed less NKp30 and NKp44 compared to lean controls. To increase the understanding on when these defects are established, we investigated NK cell frequencies and functions in a cohort of obese children aged 6–16 years of age, who were free from overt metabolic disease but do show high levels of insulin resistance. We demonstrated alterations in NK cell frequencies with reduction in NK cell number in obese children, in an insulin resistance dependent manner. Furthermore, we observed defective tumour lysis by NK cells isolated from obese children probably due to the defective production of granzyme B and perforin [67]. This suggests defects in NK cells are seeded early in obesity before the development of overt metabolic disease. Reassuringly, a study investigating the impact of weight loss following bariatric surgery on NK cells, found a normalization of NK cell cytotoxicity six months post-surgery [71]. This suggests that, with weight loss, obesity induced defects in NK cells can be reversed. This was further supported by two studies, the first of which by Jahn and colleagues showed that weight loss in obese individuals via diet and exercise resulted in increased IFN-γ production by NK cells [72]. In the second study, Barra and colleagues showed that high intensity interval training (HIIT) increased NK cell frequencies and function in obese women and mice, in addition to obesity, breast cancer cells where intravenously injected into the mice, and it was demonstrated that HIIT reduced tumour burden, with the authors postulating that this effect was mediated via increased NK cell activity (Figure 1) [73]. The processes via which weight loss or high intensity training restores NK cell functions are currently unknown.

The exact mechanisms via which obesity induces defects in NK cells are now emerging, several studies showed that leptin, an adipokine elevated in obese adults and children, can modulate NK cell functions. Leptin is a peptide hormone produced primarily by adipocytes. Its primary role is in the regulation of food intake, acting as a satiety signal [74,75,76,77]. Leptin levels are proportional to the adipose tissue mass, therefore are elevated in obese adults and children [28,78]. In addition to its role in regulation of energy homeostasis, leptin has been shown to be an important regulator of the immune system [79,80,81,82]. In leptin, receptor deficient mice (*db/db*) NK cell frequencies were reduced in the periphery, liver and spleen, suggesting that leptin is an important regulator of NK cell development [83]. In addition, the activation of NK cell was also impaired in leptin receptor deficient mice. In 2008, Nave and colleagues confirmed a role for leptin in the regulation of NK cells, and proposed a mechanism through which resistance of leptin signalling in NK cells underpinned their dysfunction in murine models of obesity [84]. Leptin resistance in obesity is a well-established feature which extends beyond the regulation of NK cells and into the classical actions of leptin in energy homeostasis [85,86]. In studies by Laue and colleagues, expression of the leptin receptor was found to be higher on NK cells from obese individuals compared to healthy controls; however, the authors noted diminished downstream signalling in obese donors. In the same study, leptin stimulation increased cytokine production (IFN-γ) in NK cells from healthy donors but not obese donors, although similar levels of proliferation were noted [87]. 

In addition to the deleterious effect of leptin resistance, we have recently elucidated another mechanism which drives defective NK cells in human obesity [22]. It is now well established that NK cell function is intrinsically dependent on cellular metabolism, with several studies highlighting the absolute requirement of NK cells to engage in glycolysis and oxidative phosphorylation [42,43,88,89]. In our recent study, we show that NK cells isolated from obese patients dysregulated metabolism, with a failure to engage glycolytic metabolism. Using RNA-sequencing, we highlighted that NK cells from obese subjects displayed elevated expression of genes involved in lipid handling and metabolism. Upon further investigation, we showed that culturing of NK cells with the free fatty acids (FFA), oleate and palmitate, recapitulated obesity-like defects in NK cells, with loss of tumour lysis, blunted cytokine production and a failure to metabolically reprogramme. Using murine models of malignancy, we show diminished anti-tumour immunity with FFA-treated NK cells, highlighting a direct link between obesity-induced NK cell defects and cancer (Figure 2) [22].

## 7. Natural Killer Cells in Adipose Tissue

In addition to their potent host protection role, NK cells also play an important role in tissue homeostasis, at sites including the uterus and adipose depots. Strikingly, Perdu and colleagues reported that maternal obesity resulted in defective NK cells in the uterus, with reduced numbers and a hyper-responsiveness resulting in increased expression of decorin which can limit trophoblast survival leading to altered placental development [68]. Recently, several studies have highlighted an important role for NK cells in adipose tissue homeostasis and the initiation of insulin resistance [24,90,91,92]. 

In 2013, O’Rourke and colleagues investigated NK cells in human adipose tissue, showing an increased activation profile in adipose tissue compared to NK cells from the blood; in a subsequent study, the authors provided for the first-time evidence that NK cells could regulate adipose tissue macrophages, suggesting a possible role in the development of insulin resistance. Systemic NK cell ablation resulted in decreased accumulation of macrophages in the adipose tissue and modest improvements in insulin sensitivity [93,94]. In 2015, Wensveen and colleagues eloquently demonstrated a role for NK cells in monitoring adipose tissue stress via the expression of natural cytotoxicity receptors (NCRs). Upon the recognition of NCRs, whose expression was upregulated on stressed adipocytes in high fat diet (HFD) fed animals, NK cells were shown to rapidly produce IFN-γ, which promoted the recruitment of macrophages into adipose tissue. These infiltrating macrophages, originally tasked with phagocytosing apoptotic adipocytes, also produced increased levels of the inflammatory cytokine IL-1β [90]. IL-1 has been extensively implicated in insulin resistance and the pathogenesis of T2DM. This pathogenic role of NK cells in insulin resistance was supported by subsequent studies from the lab of Jongson Lee, who showed that HFD increased NK cell numbers and the production of the proinflammatory cytokine TNFα, in epididymal, but not subcutaneous, adipose tissue. When NK cells were depleted, obesity-induced insulin resistance improved in parallel with decreases in both adipose tissue macrophage frequencies and inflammation. Conversely, expansion of NK cells following IL-15 administration or reconstitution of NK cells into NK cell deficient mice increased both adipose tissue macrophage infiltration and inflammation, exacerbating insulin resistance [92]. In 2017, Theurich and colleagues described a specific subset of NK cells expressing IL-6Ra and colony-stimulating factor 1 receptor (CSFR1) which were expanded in obese mice and humans and detailed their contribution to the development of insulin resistance. Ablation of CSFR1 expressing NK cell prevented the development of obesity and insulin resistance, providing further evidence of a homeostatic role for NK cell in metabolism. 

In 2016, O’Sullivan and colleagues detailed a role for adipose tissue resident innate lymphoid cells (ILC), specifically ILC1, in the development of insulin resistance [91]. NK cells are part of the ILC1 lineage, but the authors showed that ILC1 in the adipose tissue where phenotypically distinct from classical NK cells. In agreement with this, we have shown that, under steady state conditions, ILC1s are enriched in the adipose tissue from both mice and humans, and these cells display an adipose tissue specific phenotype. We show that these ILCs can regulate adipose tissue macrophages via their ability to kill inflammatory macrophages. Adipose tissue macrophages express the NKG2D ligand Rae-1, making them a target for NK cell lysis. Upon the initiation of high fat feeding, NK cells lost their ability to kill macrophages and increased their production of IFN-γ which promoted the recruitment of inflammatory macrophages, promoting obesity related metabolic defects [24]. Collectively, these studies provide strong evidence that NK cells are involved in the initiation of the macrophage-driven inflammatory phenotype reported in obese adipose tissue.

## 8. Future Directives and Open Questions

Obesity is a worldwide epidemic and is strongly associated with a milieu of chronic diseases including type II diabetes mellitus, cardiovascular disease and many cancers. NK cells represent an important front-line effector cell in host protection, including tumour surveillance and clearance. NK cell immunotherapy represents a very attractive strategy for treating cancer, in particular blood-based cancers. In this review, we have outlined the negative impact of obesity from childhood through to adulthood on NK cell function. It appears that the obese-micro environment drives defects in NK cell metabolism resulting in functional failures. As a result, there may be doubt about whether NK cell immunotherapies will be viable in obese patients. Further investigations in murine models of obesity and obese patients will be needed to definitively answer this question. In addition to their role in host protection, NK cells play an important part in the maintenance of tissues including adipose tissues. NK cells and ILC1s also directly regulate macrophages, but, with high-fat feeding, NK cells lose this ability and also alter their regulation of adipose tissue. This results in the activation and accumulation of inflammatory macrophages that drive insulin resistance—a key player in driving diabetes and cancer. Whether targeting this NK-Macrophage interaction will become an option for addressing insulin resistance and tumorigenesis remains an open question.

## Figures and Tables

**Figure 1 cancers-11-00573-f001:**
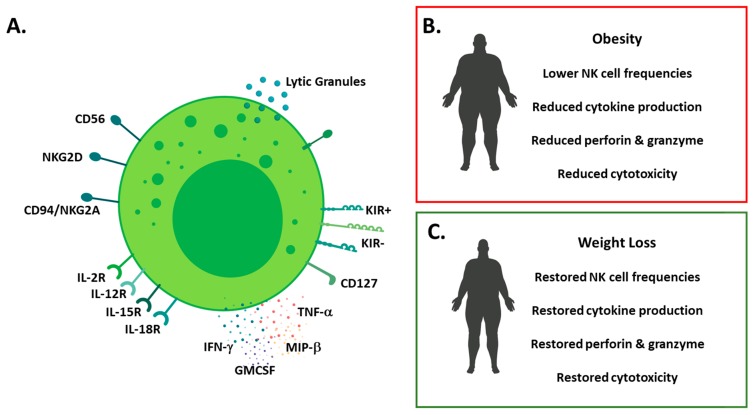
Overview of NK cell phenotype in health and obesity (**A**) schematic detailing human NK cell phenotype; surface molecules (CD56, CD16, natural killer group (NKG)2 family and killer-cell immunoglobulin-like receptors (KIRs), cytokine receptors (Interleukin(IL)-2, IL-12, IL-15 and IL-18) and effector molecules (cytokine and lytic molecule) production; (**B**) schematic detailing the impact of obesity on NK cells (lower cell frequencies, reduced cytokine production and reduced cytotoxicity); (**C**) schematic detailing the impact of weight loss on NK cells (restored frequencies, effector molecule production and cytotoxicity).

**Figure 2 cancers-11-00573-f002:**
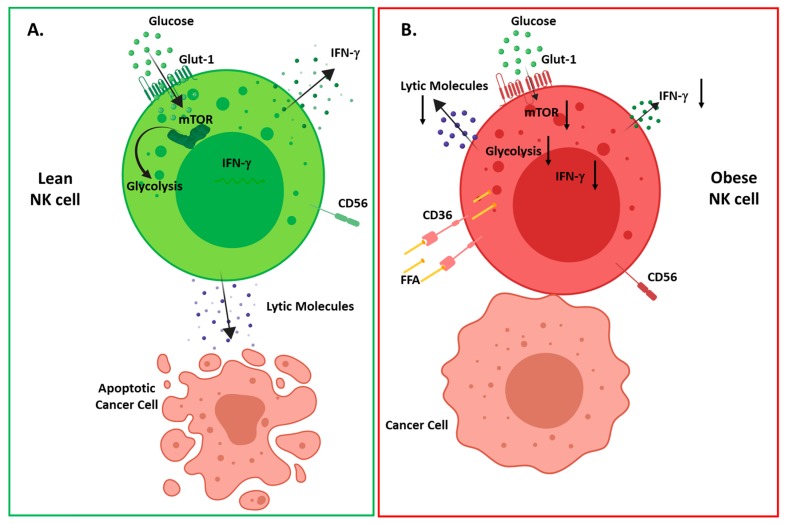
Overview of obesity induced changes in NK cell metabolism and function (**A**) in healthy individuals upon activation, NK cells increase their glycolytic metabolism mediated by mammalian target of rapamycin (mTOR), resulting in interferon (IFN)-γ production and tumor cell lysis; (**B**) in obese individuals, NK cells increase their expression of lipid uptake receptors (CD36, LDLR) resulting in increased uptake of free fatty acids. Upon activation, NK cells fail to activate mTOR and glycolytic metabolism resulting in decreased IFN-γ production, altered polarization of degranulation and defective tumor cell lysis.

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
