# Peer review of "Dysregulation of Natural Killer Cells in Obesity"

_cancers, 2019, doi:10.3390/cancers11040573_

Reviewer 1 Report

The review by O’Shea et al entitled “ Dysregulation of Natural Killer Cells in Obesity” is focusing on the role of dysregulated NK cells in obesity. Although the review is timely and novel and addresses very important subject, it falls short in providing comprehensive understanding of the mechanisms that governs dysregulation of NK cells in obesity. The authors completely overlooked the most important aspect of NK function in targeting healthy stem cells as well as cancer stem cells. This function of NK cells fully explains the mechanisms why NK cells become inactivated in obese individuals.

In addition, many recent important papers in this field have been omitted from the review, such as the studies of the NK function in Kras mutated pancreas in obese mice fed with high fat diet. Understanding the role of NK cells in targeting stem cells could clarify why NK cells become inactivated under obesity since the rate of production of adipose stem cells increases substantially necessitating control by the NK cells, however, an increase burden on the NK cells and chronic inflammatory conditions can inactivate the function of NK cells contributing to the disease process. It is unlikely that NK cells are contributing to the disease process as the authors are implying.  The authors should relate their findings to other findings within the field, otherwise their observations and interpretations of the results will be skewed and perhaps not represent the actual mechanisms that govern the functions of NK cells.

The observations regarding the loss of NK cytotoxicity in the presence of increased IFN-g (“Upon the initiation of high fat feeding, NK cells lost their ability to kill macrophages and increased their production of IFN-γ which promoted the recruitment of inflammatory macrophages, promoting obesity related metabolic defects” ) has been published extensively in the field known as “split anergy” and the rationale has been discussed in many recent reviews. The fact that such an observation is important for the induction of differentiation of the cells that NK cells interact with irrespective of the nature of the cells since NK cells do this across many different cell types including monocytes, is not due to the contributing role of NK cells in pathogenicity of the disease but it is because of the lack of proper NK cell functioning that results in the disease. In many studies it has been shown that when NK cells become defective it results in the induction of disease. The authors might not have implied the pathogenic role of NK cells but upon reading the review that is what I could infer. Please correct if that was what it was not intended.

In addition, interaction of NK cells with monocytes-macrophages, DCs and osteoclasts are very important in the activation of NK cells, therefore, this may not be a pathological event. It is when monocytes/DCs/osteoclasts lose key ligands for the activation of NK cells that the process becomes pathological. Again, it is important to cite the appropriate literature in the review and discuss their view in relation of other published reports and reviews in the field. If authors have different views it should be discussed in relation to published results.

As a minor point please spell out the abbreviations such as T2DM etc.

Also please check for any grammatical and spelling errors.

Author Response

We would like to thank reviewer 1 for their time and expertise in reviewing our manuscript. The reviewers highlights some important omissions and points which we aim to address in our revised manuscript. Below are our point by point responses to the reviewers comments.

The review by O’Shea et al entitled “ Dysregulation of Natural Killer Cells in Obesity” is focusing on the role of dysregulated NK cells in obesity. Although the review is timely and novel and addresses very important subject, it falls short in providing comprehensive understanding of the mechanisms that governs dysregulation of NK cells in obesity. The authors completely overlooked the most important aspect of NK function in targeting healthy stem cells as well as cancer stem cells. This function of NK cells fully explains the mechanisms why NK cells become inactivated in obese individuals.

We are happy that the reviewer finds our topic timely and important. We acknowledge that the important NK cell function of cancer stem cell targeting was not included in our original manuscript. We now include this in our revised manuscript and reference 4 independent studies. However we disagree with the reviewers comment that this fully explains the mechanism behind how NK cells become inactivated in obesity. We have recently published a study which highlights the obesogenic environment as one of the drivers behind NK cell dysfunction (Michelet et al, Nature Immunology 2018). In addition we discuss studies from other groups which show that leptin is an important factor in the dysregulation of NK cells in obesity, again supporting the obese environment as a mechanism. We feel this is further supported by the studies which show weight loss or exercise which change the metabolic composition of the patients results in restored NK cell function. 

In addition, many recent important papers in this field have been omitted from the review, such as the studies of the NK function in Kras mutated pancreas in obese mice fed with high fat diet. Understanding the role of NK cells in targeting stem cells could clarify why NK cells become inactivated under obesity since the rate of production of adipose stem cells increases substantially necessitating control by the NK cells, however, an increase burden on the NK cells and chronic inflammatory conditions can inactivate the function of NK cells contributing to the disease process. 

As suggested by the reviewer we now include the study which investigated NK cells in KRAS mutation mice and also include other studies which highlight cancer induced defects in NK cell functions. With respect to the proposal that increased adipose tissue stem cells result in inactivated NK cells, we have not found studies which clearly show this is the mechanism for obesity induced defects in NK cells. 

It is unlikely that NK cells are contributing to the disease process as the authors are implying.  The authors should relate their findings to other findings within the field, otherwise their observations and interpretations of the results will be skewed and perhaps not represent the actual mechanisms that govern the functions of NK cells. The observations regarding the loss of NK cytotoxicity in the presence of increased IFN-g (“Upon the initiation of high fat feeding, NK cells lost their ability to kill macrophages and increased their production of IFN-γ which promoted the recruitment of inflammatory macrophages, promoting obesity related metabolic defects” ) has been published extensively in the field known as “split anergy” and the rationale has been discussed in many recent reviews. The fact that such an observation is important for the induction of differentiation of the cells that NK cells interact with irrespective of the nature of the cells since NK cells do this across many different cell types including monocytes, is not due to the contributing role of NK cells in pathogenicity of the disease but it is because of the lack of proper NK cell functioning that results in the disease. In many studies it has been shown that when NK cells become defective it results in the induction of disease. The authors might not have implied the pathogenic role of NK cells but upon reading the review that is what I could infer. Please correct if that was what it was not intended.In addition, interaction of NK cells with monocytes-macrophages, DCs and osteoclasts are very important in the activation of NK cells, therefore, this may not be a pathological event. It is when monocytes/DCs/osteoclasts lose key ligands for the activation of NK cells that the process becomes pathological. Again, it is important to cite the appropriate literature in the review and discuss their view in relation of other published reports and reviews in the field. If authors have different views it should be discussed in relation to published results.

We provide evidence from several studies that abalation of NK cells results in improved metabolic disease (Wensveen et al, Nature Immunology 2015, O’Sullivan et al, Immunity 2916, Lee et al, Cell Metabolism 2016). Collectively these studies from independent research groups highlight a role for NK cells in the development of insulin resistance through their regulation of macrophages.

Reviewer 2 Report

This study nicely summarizes natural killer cell phenotypes in obesity. As this is a cancer journal i found the review to have too much detail on insulin resistance and not enough on cancer. This is a really hot area and a lot of biotechs are racing to develop NK-based cancer therapies (Nantkwest, Fate, etc). More details of the clinical trials mentioned would be nice. If obesity microenvironments raise doubt about the effectiveness of NK-based therapy, then is there any known exclusion criteria based on BMI made by companies in this space? All together it is a good review, but could use some more details about cancer and less about diabetes. 

Minor comments:

Line 59: other[s]

Line 75: define KIR

Line 159: do you mean deleterious?

Line 162-165: This sentence doesn't read well.

Other comments:

Use hyphens for things like obesity-induced and FFA-treated etc. There were missing commas and extra spaces each page.

Author Response

Reviewer 2

This study nicely summarizes natural killer cell phenotypes in obesity.

Firstly we would like to thank reviewer 2 for their time and expertise in reviewing our manuscript. The reviewers highlights some important points which we aim to address in our revised manuscript. Below are our point by point responses to the reviewers comments.

As this is a cancer journal I found the review to have too much detail on insulin resistance and not enough on cancer. 

We understand that our manuscript includes a section on NK cells and adipose tissue regulation, we felt it important as the NK-Obesity literature extends beyond cancer. If the editorial team and reviewers feel that we should remove this section we are happy to do so.

This is a really hot area and a lot of biotechs are racing to develop NK-based cancer therapies (Nantkwest, Fate, etc). More details of the clinical trials mentioned would be nice. If obesity microenvironments raise doubt about the effectiveness of NK-based therapy, then is there any known exclusion criteria based on BMI made by companies in this space? 

We have strived to increase the detail in the section covering NK cells and cancer therapy, however we would be keen to keep the focus on the impact of obesity on NK cells and the implication for disease. Regarding the reviewers query about BMI as exclusion criteria in NK cell immunotherapy trial, this is an excellent question. We could not find any study which definitively excluded patients due to their BMI, however many studies do exclude patients with pre-existing conditions. Some of the trials investigated are outlined below 

ClinicalTrials.gov Identifier: NCT03410368

ClinicalTrials.gov Identifier: NCT00376805

ClinicalTrials.gov Identifier: NCT03358849

ClinicalTrials.gov Identifier: NCT0121234

ClinicalTrials.gov Identifier: NCT03056339

ClinicalTrials.gov Identifier: NCT00697671

ClinicalTrials.gov Identifier: NCT02185781

ClinicalTrials.gov Identifier: NCT00640796

ClinicalTrials.gov Identifier: NCT02259348

ClinicalTrials.gov Identifier: NCT00995137

ClinicalTrials.gov Identifier: NCT00328861 

Minor comments:

Line 59: other[s]

Line 75: define KIR

Line 159: do you mean deleterious?

Line 162-165: This sentence doesn't read well.

Other comments:

Use hyphens for things like obesity-induced and FFA-treated etc. There were missing commas and extra spaces each page.

We have addressed the minor comments in our revised manuscript and would like to thank this opportunity to thank the review once again for their time and expertise.

Round  2

Reviewer 1 Report

This revised review entitled “ Dysregulation of natural killer cells in obesity” by  O’Shea and Hogan has been significantly improved by addressing all the important facts about the NK function in obesity. Several important findings in the field of NK cells has now been added and discussed in the paper which makes the review more comprehensive and detailed. In addition, the authors have now provided evidence for the loss of NK cell function in both obesity and cancer which are two very important factors synergizing in the loss of NK function as discussed by authors.

Just one slight grammatical correction in the following sentence “which promoted the of expansion of pancreatic tumours. This is supported by studies which investigated NK cell functions in patients with” Please remove of from the sentence.

Author Response

This revised review entitled “ Dysregulation of natural killer cells in obesity” by  O’Shea and Hogan has been significantly improved by addressing all the important facts about the NK function in obesity. Several important findings in the field of NK cells has now been added and discussed in the paper which makes the review more comprehensive and detailed. In addition, the authors have now provided evidence for the loss of NK cell function in both obesity and cancer which are two very important factors synergizing in the loss of NK function as discussed by authors.

We would like to thank reviewer one for their time & expertise in reviewing our manuscript, we feel the final manuscript is stronger for their input. 

Just one slight grammatical correction in the following sentence “which promoted the of expansion of pancreatic tumours. This is supported by studies which investigated NK cell functions in patients with” Please remove of from the sentence.

We have corrected this error.

Reviewer 2 Report

The edits are fine, but figure 2 legend is titled Figure 1.

Author Response

We are very happy that the reviewer is happy with our edits and would like to thank them for their time and expertise in helping refine our manuscript. We have also corrected the figure legend as highlighted by the reviewer.